# Study on the Wear Performance of Polyethylene Inner Lining Pipe under Different Load and Mineralization Conditions

Liqin Ding [1,2], Lei Wang [3,4,5,6,7,*], Jie Li [8], Suoping Qi [9], Wanli Zhang [10], Yuntao Xi [3,11], Keren Zhang [3], Shanna Xu [3], Haitao Liu [4], Lei Wen [6], Xinke Xiao [7] and Jiangtao Ji [12]

1 College of Chemistry & Chemical Engineering, Xi'an Shiyou University, Xi'an 710065, China
2 Shaanxi Key Laboratory of Carbon Dioxide Sequestration and Enhanced Oil Recovery, Xi'an 710075, China
3 School of Material Science and Engineering, Xi'an Shiyou University, Xi'an 710065, China
4 State Key Laboratory of Rolling and Automation, Northeastern University, Shenyang 110819, China
5 State Key Laboratory of Tribology in Advanced Equipment, Tsinghua University, Beijing 100084, China
6 Nation Center for Materials Service Safety, University of Science and Technology Beijing, Beijing 100083, China
7 Henan International Joint Laboratory of Dynamics of Impact and Disaster of Engineering Structures, Nanyang Institute of Technology, Nanyang 473004, China
8 Changqing Branch, China National Logging Corporation, Xi'an 710200, China
9 PipeChina West East Gas Pipeline Company, Yinchuan 750001, China
10 Sulige Gas Field Development Corporation, PetroChina Chang-Qing Oilfield Company, Xi'an 710021, China
11 Petroleum Systems Engineering, Faculty of Engineering and Applied Science, University of Regina, Regina, SK S4S 0A2, Canada
12 China Railway First Survey and Design Institute Group Co., Ltd., Xi'an 710043, China
* Correspondence: wanglei@xsyu.edu.cn or richard0723@163.com; Tel.: +86-15771921910

**Abstract:** This study conducted pin disc friction and wear performance tests on polyethylene-lined oil pipes and four types of centralizing materials (45# steel, nylon, polytetrafluoroethylene (PTFE), and surface alloy coating) in oil fields. The friction coefficient and wear rate were tested, and the wear mechanism was analyzed using scanning electron microscopy (SEM) and three-dimensional confocal microscopy. Using a combination of experimental testing analysis and theoretical research, a comprehensive evaluation of the current wellbore centering and anti-wear technology for oil was conducted. The experimental results indicate that the usage limit of polyethylene-lined oil pipes is 400 N, and compared to metal oil pipe materials, the wear rate of both stabilizing material and tubing material is lower, indicating that it has a certain service life. From the perspective of testing load, taking into account the factors of friction coefficient and wear rate, the recommended sequence of straightening material for polyethylene lined oil pipes is (1) surface alloy coating, (2) nylon, (3) PTFE, and (4) 45# steel.

**Keywords:** wear test; polyethylene inner lining pipe; straightening material; wear rate

## 1. Introduction

Directional well technology is one of the most advanced drilling technologies in the field of petroleum exploration and development in the world today [1–5]. It is a drilling technology that effectively controls the wellbore trajectory using special downhole tools, measuring instruments, and process technology, allowing the drill bit to drill in a specific direction to reach the predetermined underground target. The use of directional well technology can economically and effectively develop oil and gas resources with limited surface and underground conditions, significantly increase oil and gas production and reduce drilling costs, which is beneficial for protecting the natural environment and has significant economic and social benefits. Directional drilling is a drilling method that allows the wellbore to drill along a pre-designed wellbore inclination and orientation to reach the target layer [6].

Over 95% of the wells in Changqing Oilfield in China are developed as directional wells, leading to a continuous increase in the number of highly deviated wells in recent years. At present, the total number of oil wells produced in Changqing Oilfield is about 60,000, of which more than 20% are highly deviated wells. The wellbore trajectory of this type of well is complex and severely worn, with some wells experiencing coexistence of wear and corrosion, resulting in a decrease in oil recovery rate, frequent replacement of pipes and rods, and significant economic losses. Due to the complexity and variability of oil well trajectories, many serious problems arise, posing new challenges to traditional oil extraction methods. Among them, the problem of eccentric wear between the sucker rod and the tubing is a prominent issue in new oil wells [5,7,8]. Eccentric wear shortens the service life of the sucker rod and tubing, leading to heavy maintenance work and significantly increasing oil production costs. According to data, the maintenance frequency of equipment in Changqing Oilfield in 2020 was 42,000 times, with a maintenance cost of up to 550 million yuan. Among them, the maintenance caused by eccentric wear of sucker rods and tubing accounted for 42.8% of the total maintenance amount.

After years of rolling development, Changqing Oilfield has entered a stable production period. With the increase in liquid extraction intensity and the increase in oil well water content, the pipe and rod working conditions of pumping wells have undergone significant changes. The dynamic liquid level decreases, the pumping load increases, and the force on the pumping rod and pipe columns increases. Due to the larger diameter of the coupling of the sucker rod compared to the sucker rod itself, the coupling often comes into contact with the inner wall of the oil pipe, resulting in abrasive wear between the coupling and the oil pipe; Additionally, the wear of the coupling is much more severe than that of the sucker rod, with about 70% of the wear occurring between the coupling and the oil pipe, resulting in a decrease in coupling diameter or cracking. From the above analysis, it can be seen that there is inevitable wear and tear between the pumping rod and the oil pipe, especially in directional wells, highly deviated wells, and wells with complex wellbore trajectories. The issue of pipe and rod wear prevention in such wells has always been of great concern [9–11]. At the same time, researchers around the world have also achieved a series of important results on the issue of wear and tear [12–17].

At present, there are many anti-wear measures for pipes and pipes in Changqing Oilfield. Due to differences in blocks, mining time, production processes, and other factors, there is a lack of optimized design and suggestions on how to select and match straightening materials and pipes. Especially for the recommendation of non-metallic straightening materials for non-metallic oil pipe materials, there is an urgent need to carry out relevant research. In this current research, we compared various metallic and non-metallic straightening materials for polyethylene inner lining pipes, recommended the optimal straightening material, and determined the usage limits of different materials. At the same time, we studied the internal reasons and wear mechanisms that cause differences in wear performance between different matching materials.

## 2. Experimental

### 2.1. Materials

The tubing material in this experiment is polyethylene. The tested straightening materials include 45# steel, nylon, polytetrafluoroethylene (PTFE), paint-coated 45# steel, and cladded 45# steel, and its related parameters are shown in Table 1. The paint-coating composition is a nickel 60-coated alloy. The cladded coating is mainly composed of elements such as W, O, Ni, C, and P, with the coating mainly composed of WC and $WO_3$, as well as $Ni_2O_3$ and Ni.

**Table 1.** Analysis results of polyethylene composition.

| Type | Specifications | Compressive Strength (MPa) | Density (g/cm³) | Tensile Strength (MPa) | Elongation (%) | Molecular Formula | Molecular Mass |
|---|---|---|---|---|---|---|---|
| Polyethylene | HDPE | 24 | 0.95 | / | / | $(C_2H_4)_n$ | 100,000 |
| Polytetrafluoroethylene | 60MM | 4 | 2.2 | >15 | >150 | $CF_3(CF_2CF_2)_nCF_3$ | 100.02 |
| Nylon | PA6 | >105 | 1.14 | >90 | 20–30 | $[-NH-(CH_2)_5-CO]_n$ | 20,000 |

Hardness is usually used to represent the resistance of a material to plastic deformation caused by external indentation. For most materials, there is an approximately proportional relationship between hardness and flow stress; due to the small size of the micro-hardness testing indenter, this technique can be used to measure the hardness of materials in different regions or stages, and these results can be used as a basis for analyzing microstructure characteristics. The Rockwell hardness of 45# steel is 60, and the results of the Shore hardness test on the other collected samples are shown in Table 2.

**Table 2.** Sample microhardness test results.

| Material | Nylon | Polyethylene | Polytetrafluoroethylene | Paint-Coating |
|---|---|---|---|---|
| Shore hardness | 80 | 66 | 60 | 74 |

### 2.2. Experimental Procedure

#### 2.2.1. Friction and Wear Experiment

The pin disc friction and wear test can be used to evaluate the friction and wear performance of materials such as lubricants, metals, plastics, coatings, rubber, and ceramics. The friction pair mainly consists of point-to-surface and surface-to-surface contact friction. The friction and wear tests were carried out according to standard ASTMG99-2017. The schematic diagram and physical diagram of the pin disc friction and wear testing device are shown in Figure 1 [18,19]. In this experiment, the oil pipe material is made into a disc, and the centering material is made into a pin for a disc pin-type friction and wear experiment.

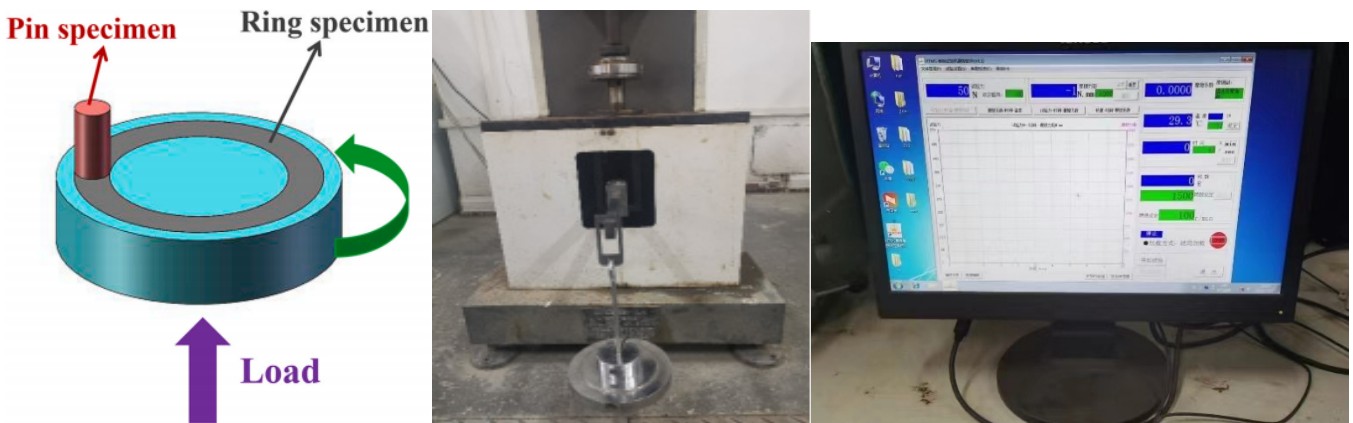

**Figure 1.** Pin disc friction and wear experimental device.

To determine the wear test time, corresponding pre-experiments were first conducted (one set of metal–non-metal and one set of metal–metal friction pairs were selected, respectively, and the wear test time was compared between 30 min and 60 min). The experiments showed that the friction coefficient tended to stabilize after 20 min of wear; After 30 min, the change can be neglected. The impact of rotational speed is relatively minimal, and there is little change in the friction and wear curves under the test conditions of 100 r/min and

200 r/min. Therefore, based on the above test results, the wear test conditions are set as follows: wear time of 30 min, friction speed of 100 r/min. Three parallel experiments were conducted for each group of friction and wear experiments to determine the repeatability of the experimental data.

The mineralization of the extracted water has a certain degree of impact on friction and wear performance. To determine the relationship between the effects, this experiment also conducted friction and wear experiments at different degrees of mineralization for each stabilizing material. In order to complete the friction and wear experiment of different mineralization degrees at an aqueous solution environment, this experiment was designed and processed in an organic glass container, as shown in Figure 2.

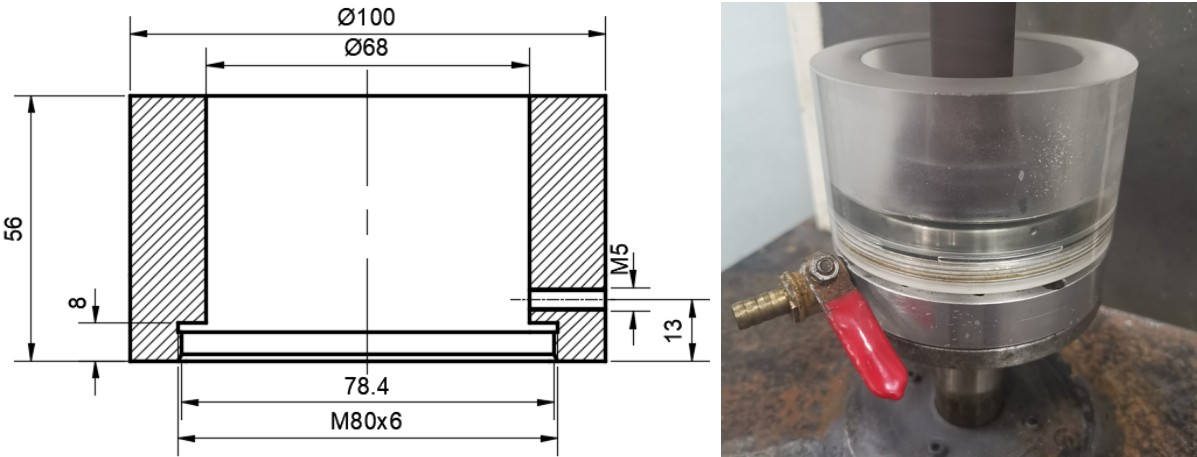

**Figure 2.** Mineralization degree aqueous solution environmental device.

### 2.2.2. Scanning Electron Microscope

With a field emission gun (Model: JSM-7001F, JEOL, Japan Electronics, Kitakyushu, Japan) operated at 20 kV, scanning electron microscopy (SEM) analyses were carried out to characterize the microstructure of worn pins and discs. The image contrast of the SEM is mainly based on the differences in the micro area characteristics of the sample surface (such as morphology, atomic number or chemical composition, crystal structure or orientation, etc.) to generate physical signals of different intensities under the action of the electron beam, resulting in different brightness differences in different areas of the cathode ray tube fluorescent screen to obtain images with a certain contrast [20,21].

### 2.2.3. Three-Dimensional Confocal Microscopy Analysis

A three-dimensional confocal microscope (Model: VHX-600E) is used to measure surface physical morphology and perform three-dimensional wear morphology analysis at micro and nano scales, such as 3D surface morphology, 2D depth morphology, contour (depth, width, curvature, angle), surface roughness.

### 3. Results

#### 3.1. The Influence of Different Environmental Media on Wear Performance

Under a load of 150 N, friction and wear experiments were conducted on polyethylene inner liner pipes and various straightening materials in aqueous solutions with different mineralization degrees. The friction coefficient curve is shown in Figure 3, and detailed information on each friction coefficient is shown in Table 3.

For polyethylene inner lining tubing materials, taking friction coefficient as the main consideration, PTFE, 45# steel, and surface alloy coating straightening materials all have excellent anti-wear effects. The priority level of the material for straightening is (1) 45# steel, (2) PTFE, and surface alloy coating, (3) nylon. For Polytetrafluoroethylene and surface alloy coatings, the friction coefficient slightly increases when the mineralization degree increases

from 80,000 mg/L to 120,000 mg/L. The friction coefficient between the polyethylene inner lining pipe and various straightening materials is relatively low but slightly higher under dry friction conditions.

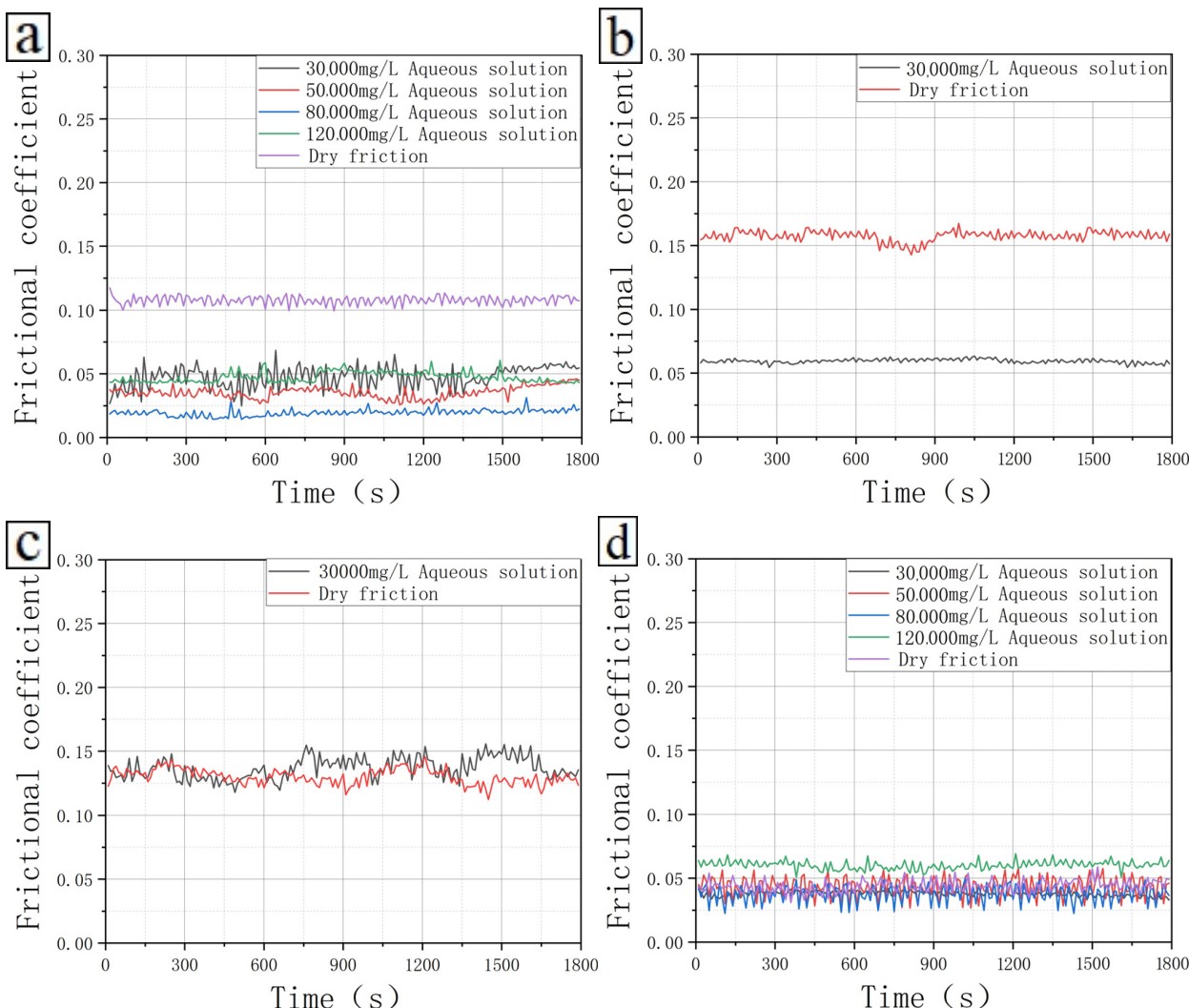

**Figure 3.** Friction coefficient of polyethylene inner lining tubing material under different mineralization degrees: (**a**) Polyethylene inner lining pipe (disc)—45# steel (pin), (**b**) Polyethylene inner lining pipe (disc)—nylon (pin), (**c**) Polyethylene inner lining pipe (disc)—PTFE (pin), and (**d**) Polyethylene inner lining pipe (disc)—surface alloy coating (pin).

**Table 3.** Friction coefficient of polyethylene inner lining tubing material under different mineralization degrees.

| Frictional Coefficient | 30,000 mg/L | 50,000 mg/L | 80,000 mg/L | 120,000 mg/L | Dry Friction |
|---|---|---|---|---|---|
| Polyethylene inner lining pipe (disc)—45# steel (pin) | 0.05 | 0.03 | 0.02 | 0.05 | 0.1 |
| Polyethylene inner lining pipe (disc)—nylon (pin) | 0.06 | / | / | / | 0.16 |
| Polyethylene inner lining pipe (disc)—PTFE (pin) | 0.13 | / | / | / | 0.12 |
| Polyethylene inner lining pipe (disc)—surface alloy coating (pin) | 0.07 | 0.06 | 0.04 | 0.05 | 0.12 |

The friction coefficient between the polyethylene inner lining tubing material and various straightening materials at different mineralization degrees is shown in Figure 4. As can be seen, under the same load conditions, the friction coefficient of dry friction of polyethylene inner lining pipes is generally significantly greater than that in solutions with different degrees of mineralization. The friction coefficient of dry friction can be increased by 20%~450% compared to that in solutions with different degrees of mineralization. Metal-containing mating pairs are more wear-resistant in aqueous solutions.

In this study, the volume lost during the wear test of the straightening material and tubing material is represented by calculating the wear rate per hundred kilometers. Tables 4 and 5 provide the wear rates of the straightening material and the tubing material per hundred kilometers in aqueous solutions with different degrees of mineralization, respectively, and their changing trends are shown in Figure 5. For polyethylene-lined oil pipes, compared to metal oil pipe materials, the wear rate of each stabilizing material is relatively low, indicating that it has a certain service life. The wear rate of each stabilizing material is comparatively high in an aqueous solution. The wear rate of the tubing material is relatively low under different straightening materials, indicating that non-metallic polyethylene lined tubing can have an excellent anti-wear effect and a longer service life. The wear rate of the friction pair composed of polyethylene-lined oil pipes and various straightening materials does not change significantly in an aqueous solution environment. From the perspective of mineralization, taking into account the factors of friction coefficient and wear rate, the recommended sequence for stabilizing materials for polyethylene-lined oil pipes are (1) polytetrafluoroethylene, (2) nylon, (3) surface alloy coating, and (4) 45# steel.

**Table 4.** Wear rate of straightening material per hundred kilometers under different mineralization degrees.

| Wear Rate per Hundred Kilometers (Straightening Material) (Unit: %) | 30,000 mg/L | 50,000 mg/L | 80,000 mg/L | 120,000 mg/L | Dry Friction |
|---|---|---|---|---|---|
| Polyethylene inner lining pipe (disc)—45# steel (pin) | 2.131 | 2.496 | 2.643 | 2.504 | 0.259 |
| Polyethylene inner lining pipe (disc)—nylon (pin) | 1.596 | 1.479 | 1.634 | 1.527 | 1.787 |
| Polyethylene inner lining pipe (disc)—PTFE (pin) | 1.274 | 1.364 | 1.524 | 1.472 | 0.987 |
| Polyethylene inner lining pipe (disc)—surface alloy coating (pin) | 7.420 | 7.869 | 7.582 | 7.677 | 0.130 |

**Table 5.** Wear rate of polyethylene lined oil pipe material per hundred kilometers.

| Wear Rate per Hundred Kilometers (Oil Pipe Material) (Unit: %) | 30,000 mg/L | 50,000 mg/L | 80,000 mg/L | 120,000 mg/L | Dry Friction |
|---|---|---|---|---|---|
| Polyethylene inner lining pipe (disc)—45# steel (pin) | 14.133 | 14.826 | 14.867 | 14.853 | 1.176 |
| Polyethylene inner lining pipe (disc)—nylon (pin) | 9.428 | 9.996 | 9.783 | 10.054 | 1.079 |
| Polyethylene inner lining pipe (disc)—PTFE (pin) | 3.243 | 3.876 | 3.548 | 3.369 | 0.975 |
| Polyethylene inner lining pipe (disc)—surface alloy coating (pin) | 4.579 | 4.978 | 4.669 | 4.823 | 1.328 |

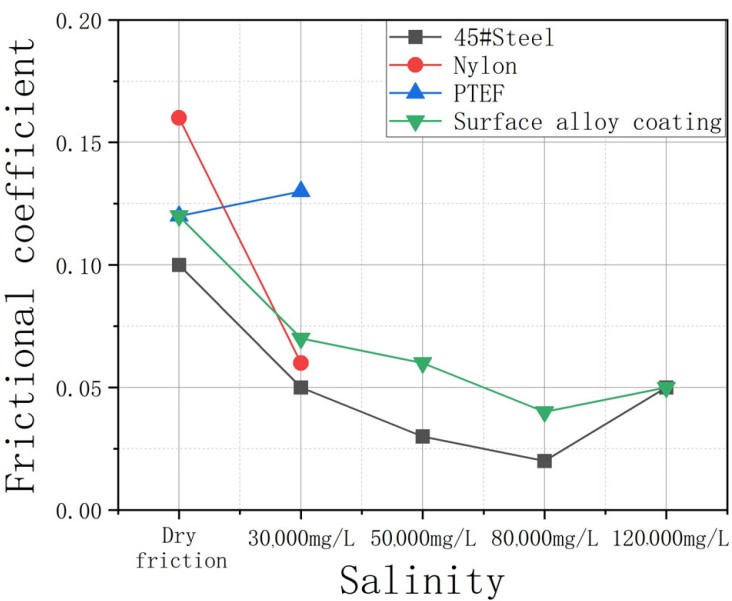

**Figure 4.** Variation of friction coefficient of polyethylene inner lining pipe with mineralization degree.

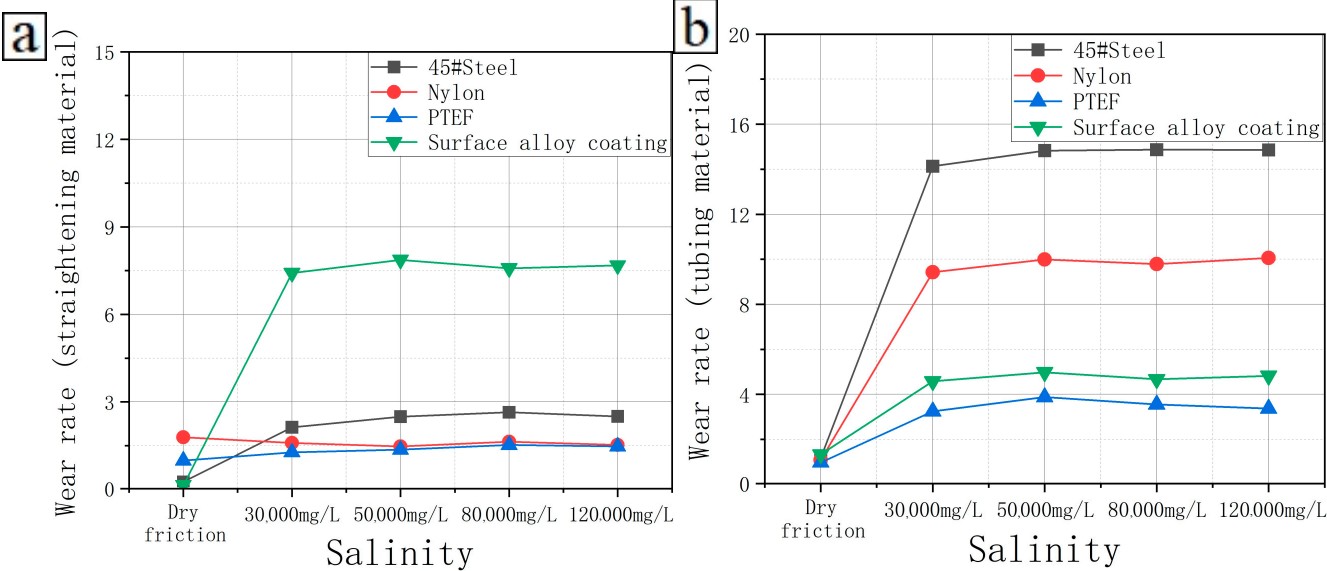

**Figure 5.** Wear rates of the straightening material and the tubing material per hundred kilometers in aqueous solutions with different degrees of mineralization. (**a**) straightening material, (**b**) tubing material.

*3.2. The Influence of Different Test Loads on Wear Performance*

3.2.1. Under Dry Friction Conditions

The friction and wear test curves of polyethylene-lined oil pipes under different loads are shown in Figure 6, and specific information on the friction coefficient is given in Table 6. Figure 7 shows the variation of friction coefficient with the load. For polyethylene-lined oil pipes, the friction coefficient of each stabilizing material does not vary significantly with the applied load. The usage limit of polyethylene inner lining pipe is 400 N. The friction coefficient between the polyethylene inner lining pipe and various straightening materials is relatively low. The usage limit of polytetrafluoroethylene (PTFE) is 150 N.

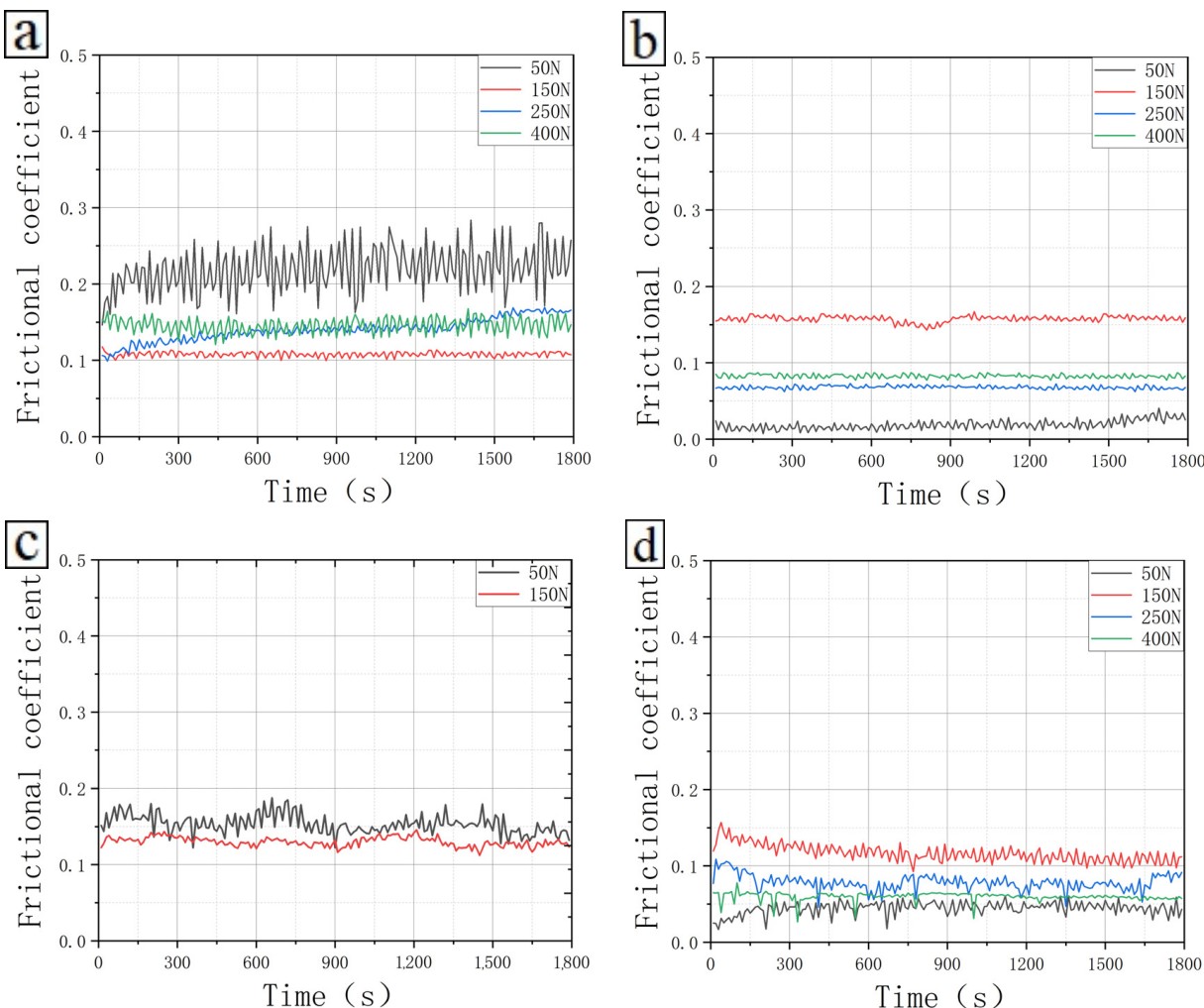

**Figure 6.** Friction coefficient of polyethylene lined oil pipe material under different test loads (dry friction): (**a**) Polyethylene inner lining pipe (disc)—45# steel (pin), (**b**) Polyethylene inner lining pipe (disc)—nylon (pin), (**c**) Polyethylene inner lining pipe (disc)—PTFE (pin), and (**d**) Polyethylene inner lining pipe (disc)—surface alloy coating (pin).

**Table 6.** Friction coefficient of polyethylene lined oil pipe material under different test loads (dry friction).

| Frictional Coefficient | 50 N | 150 N | 250 N | 400 N |
|---|---|---|---|---|
| Polyethylene inner lining pipe (disc)—45# steel (pin) | 0.21 | 0.1 | 0.15 | 0.15 |
| Polyethylene inner lining pipe (disc)—nylon (pin) | 0.02 | 0.15 | 0.07 | 0.08 |
| Polyethylene inner lining pipe (disc)—polytetrafluoroethylene (pin) | 0.16 | 0.14 | / | / |
| Polyethylene inner lining pipe (disc)—surface alloy coating (pin) | 0.05 | 0.11 | 0.08 | 0.06 |

Under different load conditions of dry friction, the wear rates of centralizing materials and tubing materials per 100 km are given in Tables 7 and 8, and their changing trends are shown in Figure 8. For polyethylene-lined oil pipes, the wear rate of each stabilizing material does not vary significantly with the applied load. Among them, nylon straightening materials have a higher wear rate and a shorter service life, which is about 1/4 of the rest of the straightening materials. The wear rate of the pipes varies significantly with the applied load when the straightening materials form a friction fit pair. Among them, the wear rate is higher, and the service life is shorter when forming a friction fit pair with 45# steel. From the perspective of testing load, taking into account the factors of friction

coefficient and wear rate, it is recommended sequentially to use (1) surface alloy coating, (2) nylon, (3) polytetrafluoroethylene, and (4) 45# steel as the straightening material for polyethylene lined oil pipes.

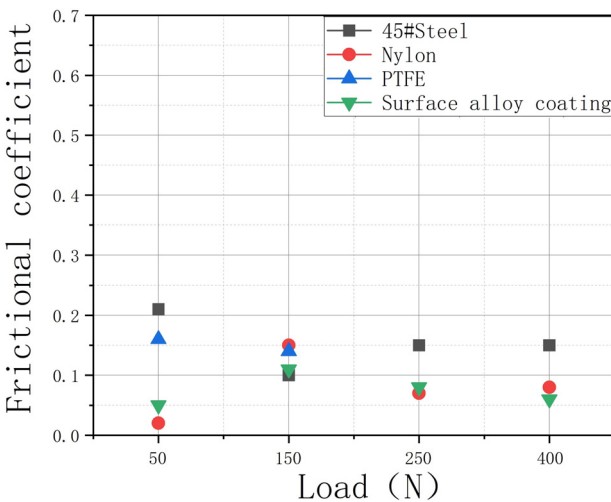

**Figure 7.** Variation of friction coefficient of polyethylene inner lining pipe with applied load (dry friction).

**Table 7.** Wear rate of straightening material per hundred kilometers under different test loads (dry friction).

| Wear Rate per Hundred Kilometers (Straightening Material) (Unit: %) | 50 N | 150 N | 250 N | 400 N |
|---|---|---|---|---|
| Polyethylene inner lining pipe (disc)—45# steel (pin) | 0.131 | 0.259 | 0.523 | 0.836 |
| Polyethylene inner lining pipe (disc)—nylon (pin) | 0.893 | 1.787 | 6.219 | 7.634 |
| Polyethylene inner lining pipe (disc)—polytetrafluoroethylene (pin) | 0.935 | 0.987 | / | / |
| Polyethylene inner lining pipe (disc)—surface alloy coating (pin) | 0.129 | 0.130 | 0.777 | 1.035 |

**Table 8.** Wear rate of oil pipe material per hundred kilometers under different test loads (dry friction).

| Wear Rate per Hundred Kilometers (Oil Pipe Material) (Unit: %) | 50 N | 150 N | 250 N | 400 N |
|---|---|---|---|---|
| Polyethylene inner lining pipe (disc)—45# steel (pin) | 0.927 | 1.176 | 2.712 | 3.787 |
| Polyethylene inner lining pipe (disc)—nylon (pin) | 0.968 | 1.079 | 1.176 | 1.341 |
| Polyethylene inner lining pipe (disc)—polytetrafluoroethylene (pin) | 0.568 | 0.975 | / | / |
| Polyethylene inner lining pipe (disc)—surface alloy coating (pin) | 0.774 | 1.328 | 1.633 | 1.934 |

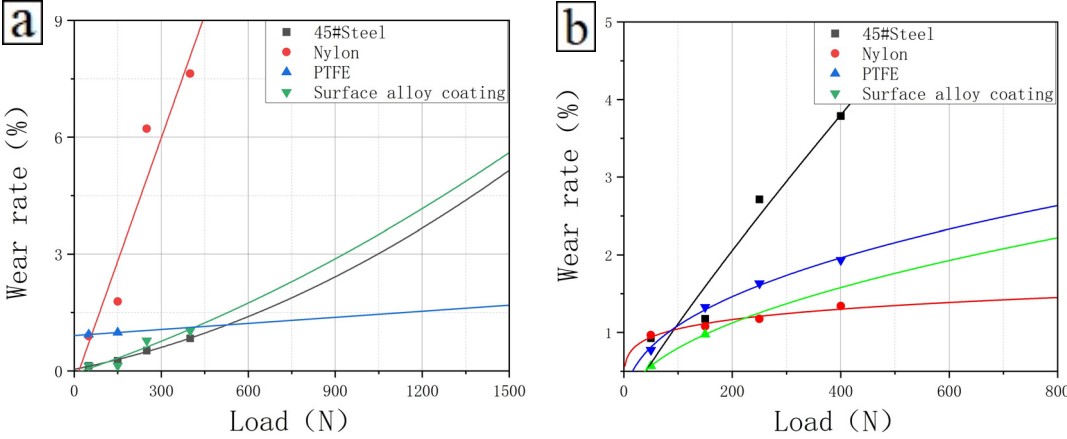

**Figure 8.** Variation of wear rate with applied load (dry friction): (**a**) straightening material and (**b**) oil pipe material.

### 3.2.2. Under Aqueous Solution Environment

To comprehensively consider the effects of mineralization degree and applied load on the friction and wear performance of polyethylene inner lining pipes, this study conducted friction and wear performance tests under different loads in a 30,000 mg/L mineralization degree aqueous solution. The friction coefficient curve is shown in Figure 9. The corresponding specific friction coefficients are given in Table 9. In addition, Tables 10 and 11 provide the wear rates per hundred kilometers of polyethylene lined oil pipe materials and straightening materials, respectively.

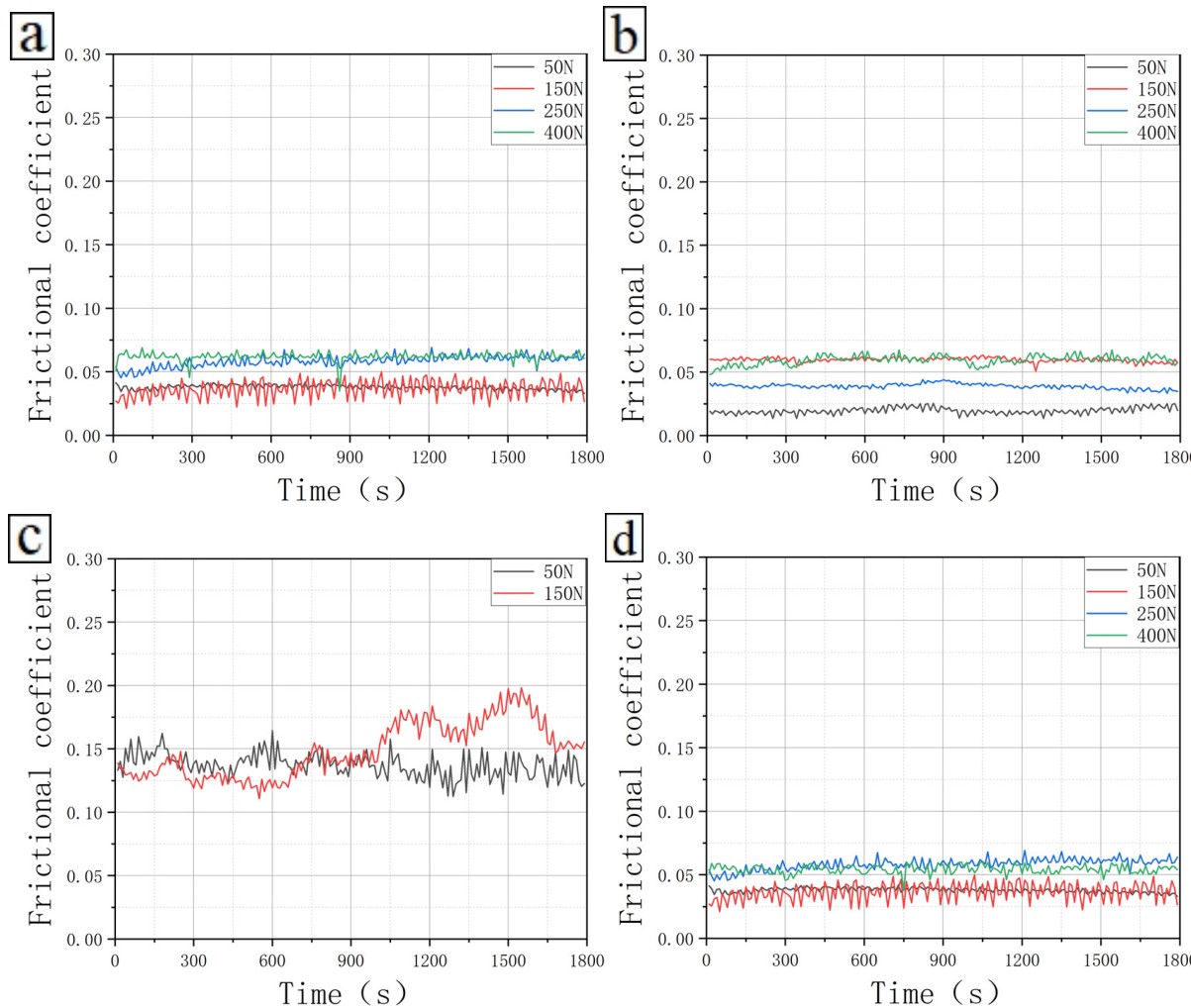

**Figure 9.** Friction coefficient of polyethylene lined oil pipe material under different test loads (aqueous solution): (**a**) Polyethylene inner lining pipe (disc)—45# steel (pin), (**b**) Polyethylene inner lining pipe (disc)—nylon (pin), (**c**) Polyethylene inner lining pipe (disc)—PTFE (pin), and (**d**) Polyethylene inner lining pipe (disc)—surface alloy coating (pin).

**Table 9.** Friction coefficient of polyethylene lined oil pipe material under different test loads (aqueous solution).

| Frictional Coefficient | 50 N | 150 N | 250 N | 400 N |
|---|---|---|---|---|
| Polyethylene inner lining pipe (disc)—45# steel (pin) | 0.04 | 0.03 | 0.05 | 0.06 |
| Polyethylene inner lining pipe (disc)—nylon (pin) | 0.02 | 0.06 | 0.04 | 0.06 |
| Polyethylene inner lining pipe (disc)—polytetrafluoroethylene (pin) | 0.15 | 0.14 | / | / |
| Polyethylene inner lining pipe (disc)—surface alloy coating (pin) | 0.03 | 0.02 | 0.05 | 0.05 |

**Table 10.** Wear rate of straightening material per hundred kilometers under different test loads (30,000 mg/L mineralization degree aqueous solution).

| Wear Rate per Hundred Kilometers (Straightening Material) (Unit: %) | 50 N | 150 N | 250 N | 400 N |
|---|---|---|---|---|
| Polyethylene inner lining pipe (disc)—45# steel (pin) | 0.124 | 0.147 | 0.409 | 0.694 |
| Polyethylene inner lining pipe (disc)—nylon (pin) | 0.798 | 1.620 | 6.044 | 7.105 |
| Polyethylene inner lining pipe (disc)—polytetrafluoroethylene (pin) | 0.779 | 0.872 | / | / |
| Polyethylene inner lining pipe (disc)—surface alloy coating (pin) | 0.127 | 0.129 | 0.694 | 0.925 |

**Table 11.** Wear rate of oil pipe material per hundred kilometers under different test loads (30,000 mg/L mineralization degree aqueous solution).

| Wear Rate per Hundred Kilometers (Oil Pipe Material) (Unit: %) | 50 N | 150 N | 250 N | 400 N |
|---|---|---|---|---|
| Polyethylene inner lining pipe (disc)—45# steel (pin) | 0.872 | 1.013 | 2.521 | 3.462 |
| Polyethylene inner lining pipe (disc)—nylon (pin) | 0.761 | 0.972 | 0.991 | 1.136 |
| Polyethylene inner lining pipe (disc)—polytetrafluoroethylene (pin) | 0.524 | 0.861 | / | / |
| Polyethylene inner lining pipe (disc)—surface alloy coating (pin) | 0.679 | 1.156 | 1.346 | 1.721 |

From the variation of friction coefficient with applied load in the aqueous solution environment shown in Figure 10, it can be seen that for polyethylene-lined oil pipes, the friction coefficient of each stabilizing material does not change significantly with the applied load. The friction coefficient between the polyethylene inner lining pipe and various straightening materials is low, among which the friction coefficient with PTFE is higher.

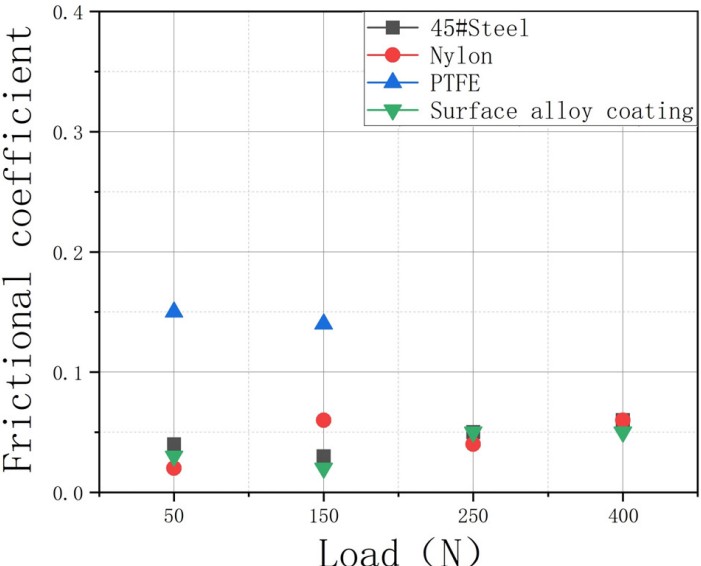

**Figure 10.** Variation of friction coefficient of polyethylene inner lining pipe with applied load (30,000 mg/L mineralization degree aqueous solution).

In terms of the wear rate of the straightening material and oil pipe material, the trend of variation is extremely similar to the wear rate under dry friction conditions, as shown in Figure 11. Specifically, the wear rate of nylon straightening material is relatively high. When it forms a friction fit pair with 45# steel, the wear rate of polyethylene lined oil pipe material is relatively high, indicating a lower service life in both cases.

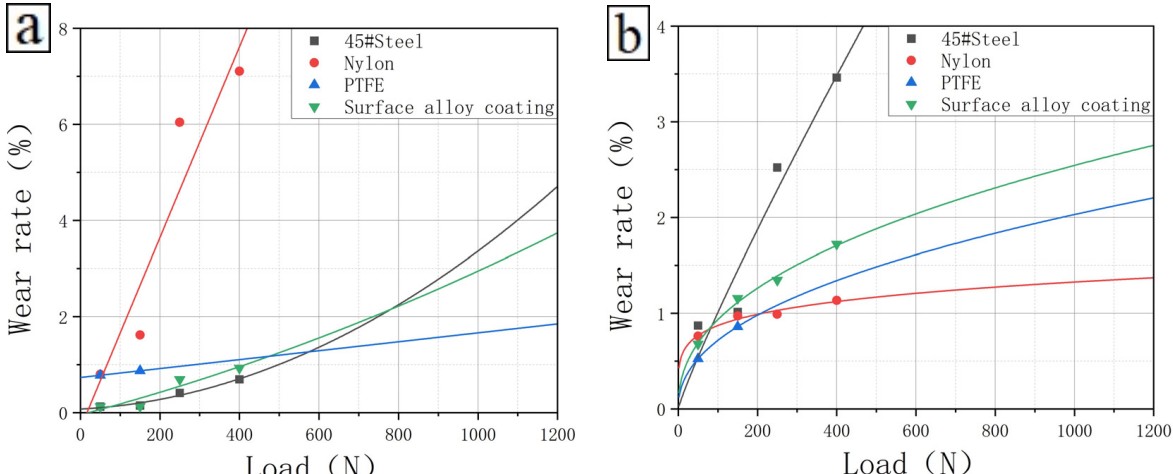

**Figure 11.** Variation of wear rate with applied load (dry friction): (**a**) straightening material and (**b**) oil pipe material (30,000 mg/L mineralization degree aqueous solution).

### 3.3. Analysis of Wear Mechanism

From the above test results, it can be seen that the polyethylene inner lining pipe and four types of straightening materials (45# steel, nylon, polytetrafluoroethylene, surface alloy coating) have lower friction coefficients in different mineralization aqueous solutions and under different loads. In order to reveal the wear mechanism of this low friction coefficient pair, SEM analysis, and three-dimensional confocal microscopy analysis were performed on the polyethylene (disc)–surface alloy coating (pin) under different loads in a 30,000 mg/L mineralization degree aqueous solution. The results are shown in Figures 12–14.

By analyzing the surface SEM microstructure of the worn disc and pin, it can be seen that the friction coefficient of the polyethylene inner lining tube (disc)–surface alloy coating (pin) friction pair is relatively low, which is recommended as a friction pair. It can be seen that there are a small amount of furrows parallel to the sliding direction, and there are a small amount of white particles on the surface, which belong to lighter particle wear, and no obvious furrows are observed. The three-dimensional confocal microscopy analysis also shows that the friction coefficient of the polyethylene inner lining tube (disc)–surface alloy coating (pin) friction pair is low, which is a typical low friction coefficient low wear rate pair. The three-dimensional confocal experiment shows that there are obvious furrows with small and uniform spacing under various loads [22–24].

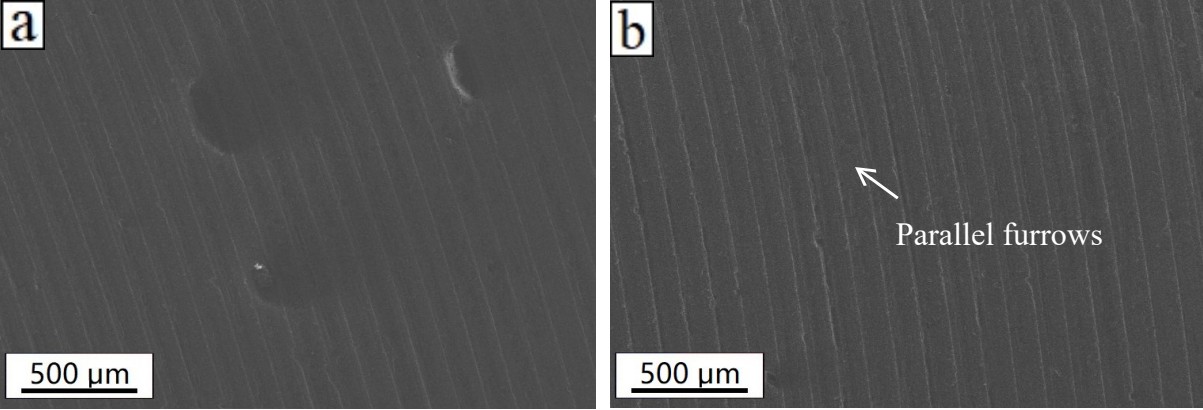

**Figure 12.** *Cont.*

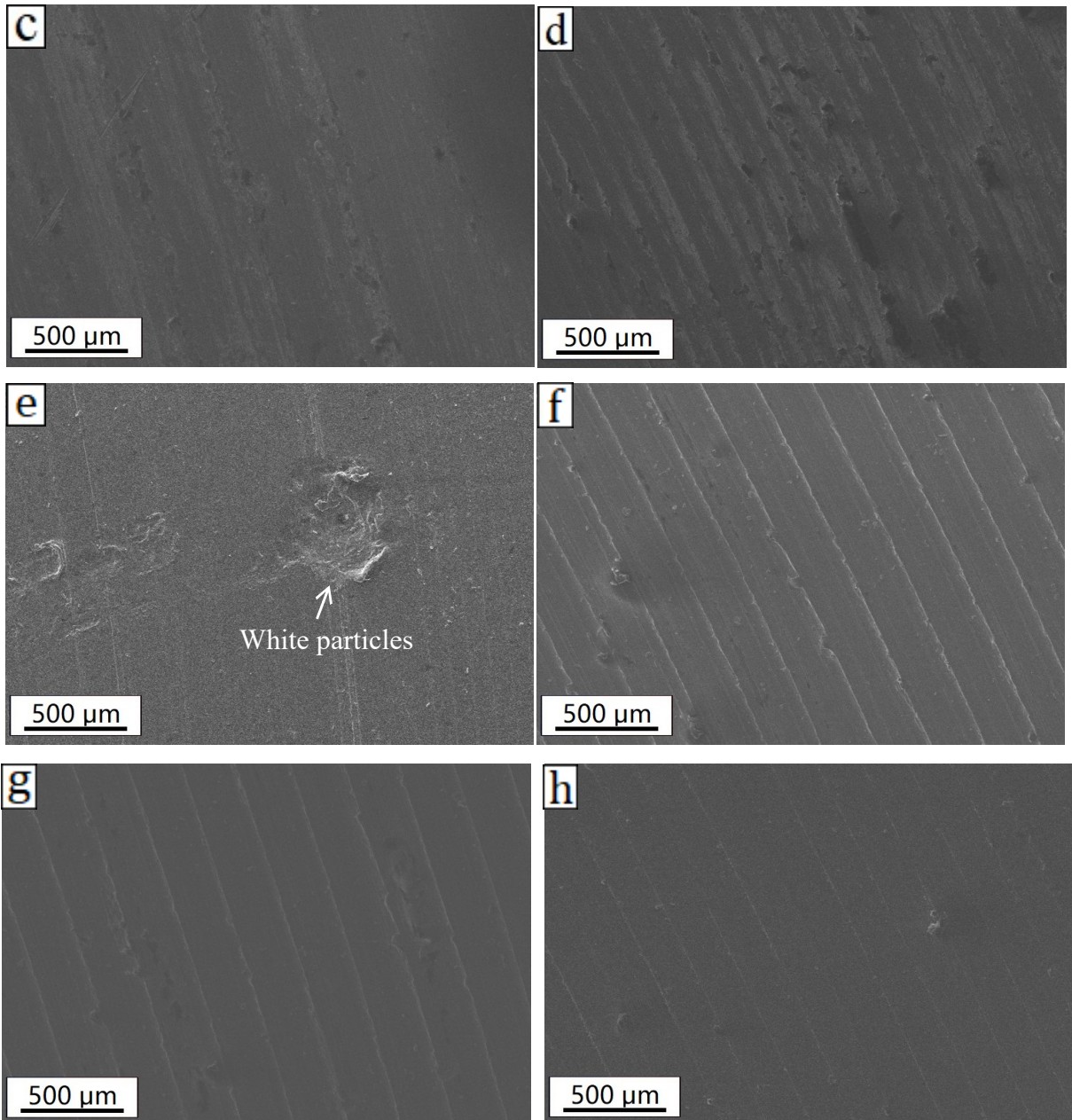

**Figure 12.** SEM images of the worn surface of polyethylene inner lining pipe (disc)–surface alloy coating (pin) under different loading conditions in a 30,000 mg/L mineralization degree aqueous solution: (**a**) surface alloy coatings under 50 N, (**b**) polyethylene under 50 N, (**c**) surface alloy coatings under 150 N, (**d**) polyethylene under 150 N, (**e**) surface alloy coatings under 250 N, (**f**) polyethylene under 250 N, (**g**) surface alloy coatings under 400 N, and (**h**) polyethylene under 400 N.

Under the corrosion of highly mineralized water, the wear rate of the material is lower than that under dry friction conditions, and the wear mechanism is a combination of corrosive wear and abrasive wear. At low loads, the lubrication of liquid media plays a major role, and at this time, material loss is determined by corrosion behavior [25–27]. When the load increases, the surface of the material exhibits elastic contact, and the loss of the metal material is still caused by corrosion. As the load continues to increase, the surface of the metal will reach a state of plastic contact, and the material loss is caused by the interaction of corrosion and wear [28].

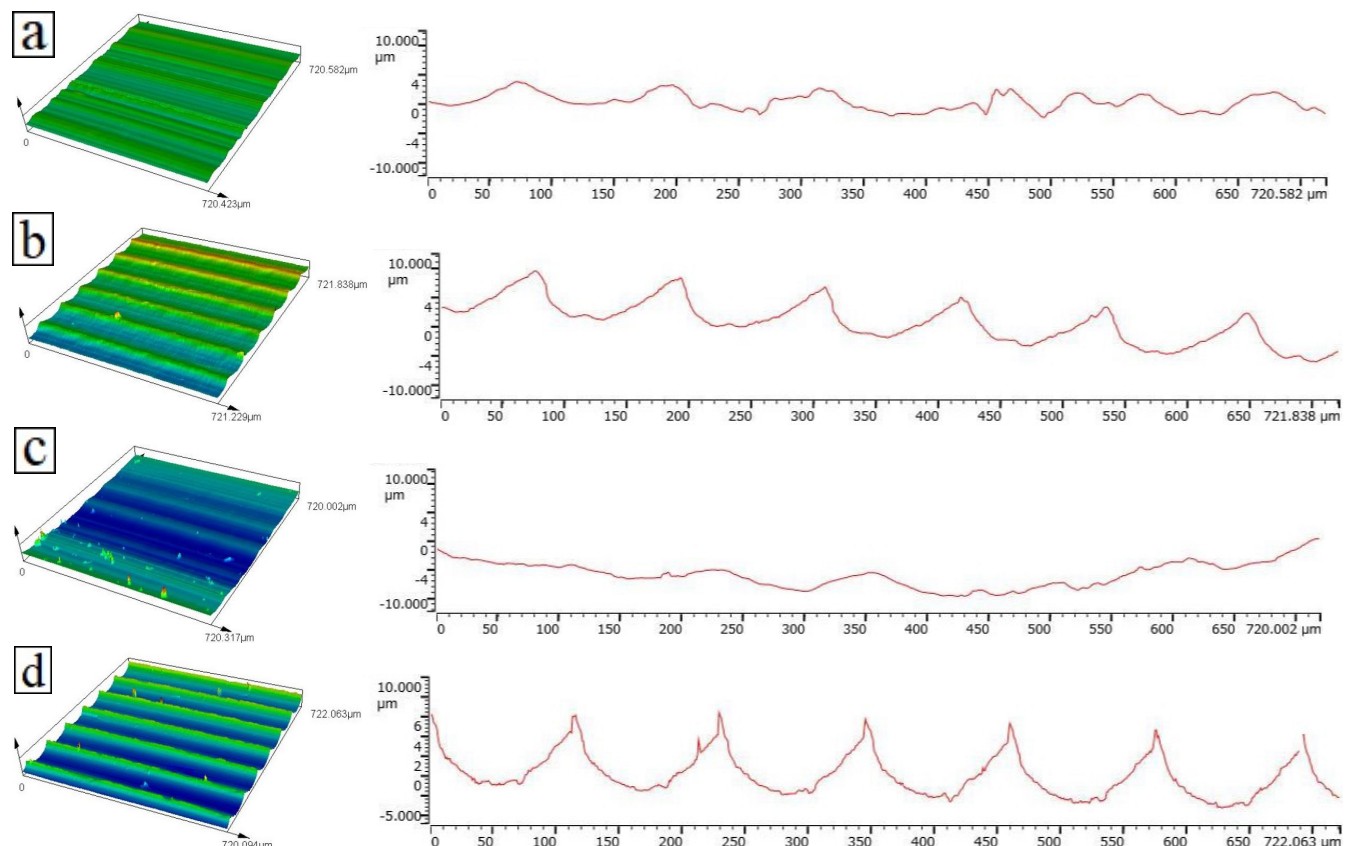

**Figure 13.** Three—Dimensional Confocal Microscopic Images and height contour of polyethylene inner lining pipe (disc)—surface alloy coating (pin) under different loading conditions in a 30,000 mg/L mineralization degree aqueous solution: (**a**) polyethylene under 50 N, (**b**) polyethylene under 150 N, (**c**) polyethylene under 250 N, and (**d**) polyethylene under 400 N.

Under the same load conditions, the friction coefficient and wear rate of dry friction are generally significantly higher than those in solutions with different mineralization degrees. The wear microstructure under various loads in an aqueous solution is similar to that under dry friction conditions, but the friction coefficient and wear rate are lower than those under dry friction conditions. When the wear is severe (such as when the load is greater than 400 N), the wear mechanism is mainly close to adhesive wear, and the furrow is generally deep, up to tens of micrometers, and the distribution is uneven. When the wear is slight (such as when the load is less than 200 N), the furrow is generally shallow, generally within 10 microns, and evenly distributed [29–31].

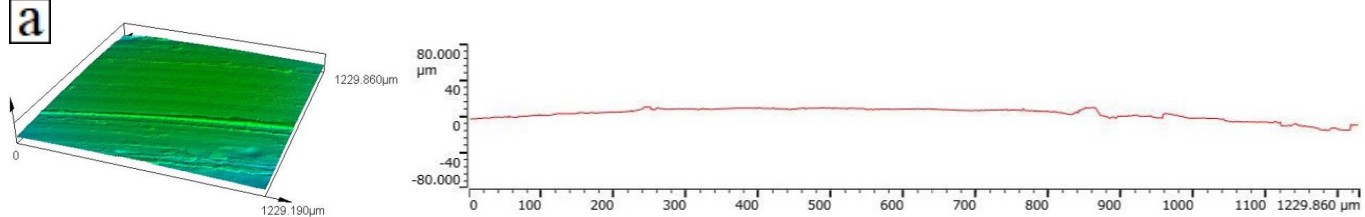

**Figure 14.** *Cont.*

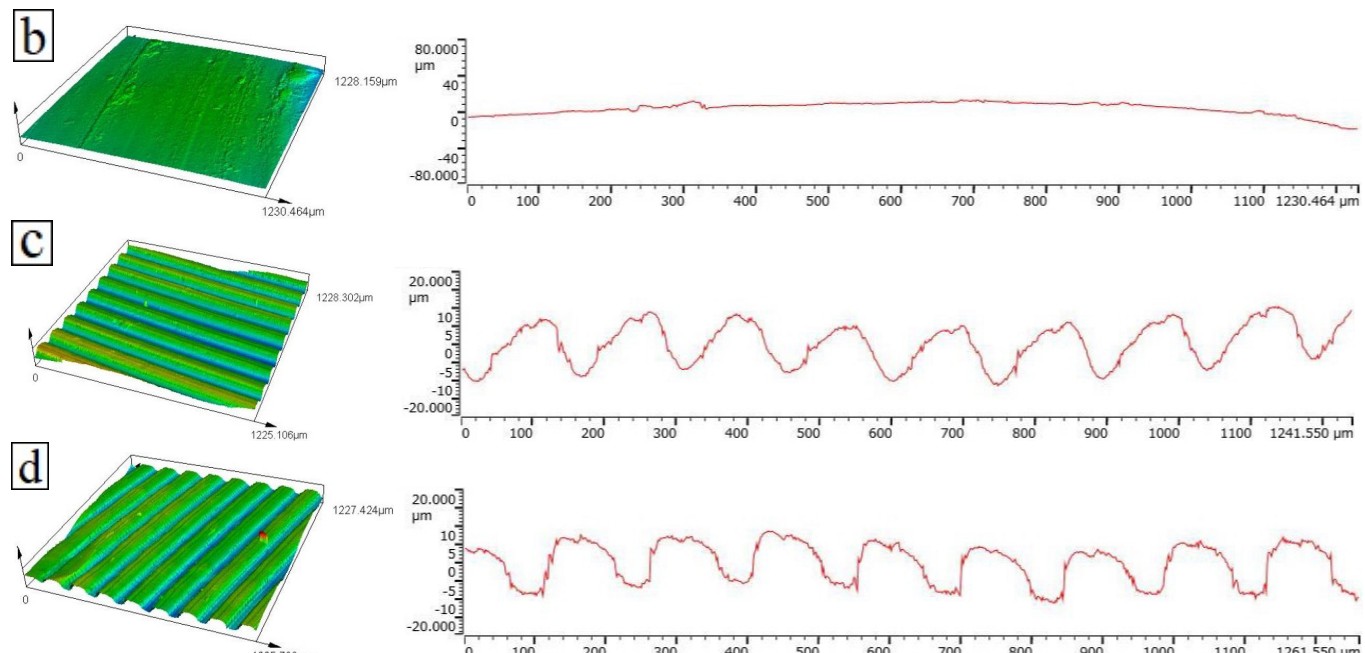

**Figure 14.** Three—Dimensional Confocal Microscopic Images and height contour of polyethylene inner lining pipe (disc)—surface alloy coating (pin) under different loading conditions in a 30,000 mg/L mineralization degree aqueous solution: (**a**) surface alloy coatings under 50 N, (**b**) surface alloy coatings under 150 N, (**c**) surface alloy coatings under 250 N, and (**d**) surface alloy coatings under 400 N.

## 4. Conclusions

In this experiment, friction and wear performance tests were conducted on four types of straightening materials for polyethylene-lined oil pipes under different mineralization degrees and load conditions. The friction coefficient, wear rate, and wear mechanism under different conditions were analyzed. The main conclusions obtained are as follows:

1.  Polyethylene-lined oil pipes, compared to metal oil pipe materials, have lower friction coefficients which are generally below 0.2 with various straightening materials;
2.  For polyethylene-lined oil pipes, compared to metal oil pipe materials, the wear rate of both stabilizing material and tubing material is lower, indicating that it has a longer service life;
3.  From the perspective of testing load, taking into account the factors of friction coefficient and wear rate, the recommended sequence of straightening material for polyethylene lined oil pipes is (1) surface alloy coating, (2) nylon, (3) PTFE, and (4) 45# steel.

**Author Contributions:** Methodology, L.D.; Validation, L.W. (Lei Wang) and W.Z.; Investigation, S.Q. and J.J.; Resources, H.L. and X.X.; Data curation, J.L. and K.Z.; Writing—original draft, L.W. (Lei Wang); Writing—review & editing, Y.X. and S.X.; Project administration, L.W. (Lei Wen). All authors have read and agreed to the published version of the manuscript.

**Funding:** This present work has been financially supported by National Natural Science Foundation of China Project (Grant No.: 12102340), Open Foundation of Shaanxi Key Laboratory of Carbon Dioxide Sequestration and Enhanced Oil Recovery (YJSYZX23SKF0007), Natural Science Basic Research Program of Shaanxi Province (Program No. 2022JM-078), Young Scientific Research and Innovation Team of the Xi'an Shiyou University (Grant No.: 2019QNKYCXTD14), The Open Research Fund from the State Key Laboratory of Rolling and Automation, Northeastern University (Grant No.: 2022RALK-FKT009), The Tribology Science Fund of State Key Laboratory of Tribology in Advanced Equipment (Grant No.: SKLTKF22B10), State Key Lab of Advanced Metals and Materials (Grant No.: 2021-Z06), Opening project fund of Materials Service Safety Assessment Facilities (Grant No.: MSAF-2021-101),

**Institutional Review Board Statement:** Not applicable.

**Informed Consent Statement:** Not applicable.

**Data Availability Statement:** Not applicable.

**Acknowledgments:** This present work has been financially supported by National Natural Science Foundation of China Project (Grant No.: 12102340), Open Foundation of Shaanxi Key Laboratory of Carbon Dioxide Sequestration and Enhanced Oil Recovery (YJSYZX23SKF0007), Natural Science Basic Research Program of Shaanxi Province (Program No. 2022JM-078), Young Scientific Research and Innovation Team of the Xi'an Shiyou University (Grant No.: 2019QNKYCXTD14), The Open Research Fund from the State Key Laboratory of Rolling and Automation, Northeastern University (Grant No.: 2022RALKFKT009), The Tribology Science Fund of State Key Laboratory of Tribology in Advanced Equipment (Grant No.: SKLTKF22B10), State Key Lab of Advanced Metals and Materials (Grant No.: 2021-Z06), Opening project fund of Materials Service Safety Assessment Facilities (Grant No.: MSAF-2021-101), Henan International Joint Laboratory of Dynamics of Impact and Disaster of Engineering Structures, Nanyang Institute of Technology (Grant No.: LDIDES-KF2022-02-02), and China Scholarship Council Foundation (Grant No.: 202208615046).

**Conflicts of Interest:** The authors declare no conflict of interest.

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
