# Peer review of "Study on the Wear Performance of Polyethylene Inner Lining Pipe under Different Load and Mineralization Conditions"

_coatings, doi:10.3390/coatings13071155_

Round 1

Reviewer 1 Report

The paper undertakes a study on the wear performance of polyethylene inner lining pipe under different load and mineralization conditions. 

The research results are presented in an interesting way.

Please:

- verify the article linguistically,

- emphasize the novelty of research in relation to other research papers,

- calculate measurement errors,

- determine the repeatability of the measurement results.

Please verify the article linguistically.

Author Response

Editor of Coatings.

Electronic submission

Xi′an, May 23th 2023

Dear Editor:

Please find enclosed the revised manuscript entitled “Study on the Wear Performance of Polyethylene Inner Lining Pipe under Different Load and Mineralization Conditions” with reference coatings-2418198. The changes with respect the former text are in yellow colour. Moreover, in the next pages of this letter the “response to reviewers” is included.

We hope that you find the paper suitable for publication in your Journal.

Looking forward to hearing from you soon,

Yours sincerely

Dr. Wang Lei

School of Materials Science and Engineering

Xian Shiyou University, Xi′an, shaanxi, 710065, PR China

Tel. Phone: (86)-15771921910

E-mail address: wanglei@xsyu.edu.cn, richard0723@163.com

The paper undertakes a study on the wear performance of polyethylene inner lining pipe under different load and mineralization conditions. 

The research results are presented in an interesting way.

Please:

  • Verify the article linguistically.

Yes, we have checked the English expressions and corrected grammatical error in the manuscript.

  • Emphasize the novelty of research in relation to other research papers.

We have added relevant explanations in the manuscript.

  • Calculate measurement errors.

The measurement of friction coefficient is directly obtained through experimental instruments, and the wear rate is calculated, so there is no indication of error in the manuscript.

  • Determine the repeatability of the measurement results.

We have explained the repeatability of the data in the text.

Reviewer 2 Report

In general, the topic addressed in the paper Study on the Wear Performance of polyethylene Inner Lining Pipe under Different Load and Mineralization Conditions” by W. Zhang, et al. is very interesting. The paper shows novel results in the field of line oil pipes. Meanwhile the paper needs minor changes in order to be suitable for publication in Coatings.

Please find below a list of observations:

1. In the list of authors appears “and”, it is convenient to delete this.

2. Lines 26, 91 and 184: write “(PTFE)” after the term polytetrafluoroethylene.

3. Line 32: delete the words “For polyethylene lined oil pipes” since they were written in the previous line, 31.

4. Line 149: what does “>PTFE=surface alloy coating”

5. Lines 151, 200 and 220: separate by comma symbol the quantities 80,000; 120,000; as well as 30,000.

6.  Line 225: It is convenient to show the surface characteristics seen by SEM by using geometrical shapes, including arrows, circles, etc. For example, usage of arrows for to show the “small amount of furrows parallel to the sliding direction”.

7. In Table 1: Correct “Mpa” by “MPa”. Write the units of each parameter next to it, for example, “Density (g/cm3)”; “Elongation (%)”, and delete the units in each result. Separate the values by commas in molecular mass results.

8.  Figures 7 and 10:  Write in the axis X “Load (N)”, and delete “N” in each term in the axis.

Author Response

Editor of Coatings.

Electronic submission

Xi′an, May 23th 2023

Dear Editor:

Please find enclosed the revised manuscript entitled “Study on the Wear Performance of Polyethylene Inner Lining Pipe under Different Load and Mineralization Conditions” with reference coatings-2418198. The changes with respect the former text are in yellow colour. Moreover, in the next pages of this letter the “response to reviewers” is included.

We hope that you find the paper suitable for publication in your Journal.

Looking forward to hearing from you soon,

Yours sincerely

Dr. Wang Lei

School of Materials Science and Engineering

Xian Shiyou University, Xi′an, shaanxi, 710065, PR China

Tel. Phone: (86)-15771921910

E-mail address: wanglei@xsyu.edu.cn, richard0723@163.com

In general, the topic addressed in the paper “Study on the Wear Performance of polyethylene Inner Lining Pipe under Different Load and Mineralization Conditions” by W. Zhang, et al. is very interesting. The paper shows novel results in the field of line oil pipes. Meanwhile the paper needs minor changes in order to be suitable for publication in Coatings.

Please find below a list of observations:

  1. In the list of authors appears “and”, it is convenient to delete this.

We have deleted it.

  1. Lines 26, 91 and 184: write “(PTFE)” after the term polytetrafluoroethylene.

We have made modifications.

  1. Line 32: delete the words “For polyethylene lined oil pipes” since they were written in the previous line, 31.

We have deleted it.

  1. Line 149: what does “>PTFE=surface alloy coating”

We have made modifications to this sentence.

  1. Lines 151, 200 and 220: separate by comma symbol the quantities 80,000; 120,000; as well as 30,000.

We have made modifications.

  1. Line 225: It is convenient to show the surface characteristics seen by SEM by using geometrical shapes, including arrows, circles, etc. For example, usage of arrows for to show the “small amount of furrows parallel to the sliding direction”.

We have added the content in the figure.

  1. In Table 1: Correct “Mpa” by “MPa”. Write the units of each parameter next to it, for example, “Density (g/cm3)”; “Elongation (%)”, and delete the units in each result. Separate the values by commas in molecular mass results.

We have made modifications to the table.

  1. Figures 7 and 10:  Write in the axis X “Load (N)”, and delete “N” in each term in the axis.

We have made modifications to the figure.

Reviewer 3 Report

Comments to authors:

This paper “Study on the Wear Performance of Polyethylene Inner Lining 2 Pipe under Different Load and Mineralization Conditions”. I found the work to be severely lacking in content and have recommended major corrections. My specific comments are included below, in no particular sequence.

Suggestion to authors

1)            The abstract could offer more of a design guide to developers.

The introduction might be improved by citing more recent research on the study's aims and objectives.

2)            The paper needs improvement in its structure. It also needs to be more decisive in the argument it is presenting.

3)            Experimental methods need to be rewritten and cited from literature work.

4)            The methodology does not mention any standard was used in this study.

5)            A schematic diagram of the experimental setup is required for methodology.

6)            The work's limitations and implications should be discussed more clearly.

7)            Seek to compare the experimental results with earlier work to see which is better.

8)            The result and discussion were poorly written. To justify the findings, new citations, and debate are needed.

9)            Quality of the Figures should be improved.

10)        Conclusion: Please revise this part as well as this conclusion part needs to conclude every result that have been obtained in a precise and understandable manner so that the viewers could understand well the overall findings that have been obtained.

Moderate editing of English language required

Author Response

Editor of Coatings.

Electronic submission

Xi′an, June 2ed 2023

Dear Editor:

Please find enclosed the revised manuscript entitled “Study on the Wear Performance of Polyethylene Inner Lining Pipe under Different Load and Mineralization Conditions” with reference coatings-2418198. The changes with respect the former text are in yellow colour. Moreover, in the next pages of this letter the “response to reviewers” is included.

We hope that you find the paper suitable for publication in your Journal.

Looking forward to hearing from you soon,

Yours sincerely

Dr. Wang Lei

School of Materials Science and Engineering

Xian Shiyou University, Xi′an, shaanxi, 710065, PR China

Tel. Phone: (86)-15771921910

E-mail address: wanglei@xsyu.edu.cn, richard0723@163.com

This paper “Study on the Wear Performance of Polyethylene Inner Lining 2 Pipe under Different Load and Mineralization Conditions”. I found the work to be severely lacking in content and have recommended major corrections. My specific comments are included below, in no particular sequence.

Suggestion to authors

1)  The abstract could offer more of a design guide to developers.

The introduction might be improved by citing more recent research on the study's aims and objectives.

Yes, we have made modifications to the abstract and introduction sections, and added relevant references.

2)  The paper needs improvement in its structure. It also needs to be more decisive in the argument it is presenting.

We have analyzed the structure of the manuscript and modified the structure. Now it has been divided it into three aspects: (1) the impact of environmental media on wear performance, (2) the impact of applied load on wear performance, and (3) wear mechanism. We find this is reasonable.

3)  Experimental methods need to be rewritten and cited from literature work.

We added references in the experimental methods parts and rewrote some sentences.

4)  The methodology does not mention any standard was used in this study.

We have added the execution standards for friction and wear tests in this study.

5)  A schematic diagram of the experimental setup is required for methodology.

The schematic diagram of the experimental setup has been added in Figure 1.

6)  The work's limitations and implications should be discussed more clearly.

We have added relevant statements on the main limitations and implications of this work.

7)  Seek to compare the experimental results with earlier work to see which is better.

This manuscript mainly compares and analyzes the wear performance of polyethylene lined oil pipes and metal oil pipes on different straightening materials. However, due to different conditions, the comparison of the wear performance of polyethylene inner lining pipes before work was not included in this manuscript.

8)  The result and discussion were poorly written. To justify the findings, new citations, and debate are needed.

We have added references and relevant expressions in the results and discussion section.

9)  Quality of the Figures should be improved.

We have replaced the figures with poor quality.

10)  Conclusion: Please revise this part as well as this conclusion part needs to conclude every result that have been obtained in a precise and understandable manner so that the viewers could understand well the overall findings that have been obtained.

We have made modifications to the conclusion section to make the conclusion clearer to readers.
